# Corrosion Resistance in Artificial Perspiration of Cr-Based Decorative Coatings

**DOI:** 10.3390/nano13162346

**Published:** 2023-08-15

**Authors:** Edgar Carneiro, José David Castro, Maria José Lima, Jorge Ferreira, Sandra Carvalho

**Affiliations:** 1CF-UM-UP-Physics Center of Minho and Porto Universities, Physics Department, University of Minho, 4800-058 Guimarães, Portugal; mjlima@fisica.uminho.pt; 2CEMMPRE-Centre for Mechanical Engineering, Materials and Processes (CEMMPRE), Department of Mechanical Engineering, University of Coimbra, 3030-788 Coimbra, Portugal; uc2021120076@student.uc.pt; 3Engineering Department, KLC—Technical Plastics, 2430-021 Marinha Grande, Portugal; jorge.ferreira@klc.pt; 4LED & MAT-IPN—Laboratory for Wear, Testing and Materials, Instituto Pedro Nunes, Rua Pedro Nunes, 3030-199 Coimbra, Portugal

**Keywords:** aluminum, chromium oxide, chromium nitride, sputtered films, EIS, corrosion

## Abstract

We aim at developing hexavalent chromium-free coatings for frequently touched decorative parts. Cr(N,O) and multilayered CrN/CrO coatings were deposited by means of reactive magnetron sputtering. All samples presented good adhesion to the substrates enhanced by an epoxy layer designed to enhance PVD coating adhesion. Similar substrates are found in the automotive industry and can be used in appliances where a metallic finish is desired by the consumer. Corrosion behavior was induced, using artificial sweat to simulate long exposure to human touch for 96 h. In potentiodynamic polarization tests, the coatings were revealed to be nobler than the substrate alone. Cr displayed a non-existent passivation region, while gCrN exhibited a quick passivation of the surface and its respective breakdown and several current fluctuations, indicating the occurrence of pitting, which was confirmed by SEM micrography after the corrosion. Regarding EIS results, all films depicted a diminution of impedance modulus (|Z|) after 96 h, which indicates a diminution of corrosion resistance against artificial sweat. Nitride films exhibited the worst anticorrosive features. On the other hand, Cr and CrO exhibited the highest |Z| values. These results are corroborated by low the corrosion rates of both coatings. The equivalent electrical circuit allows us to confirm oxide formation in the outermost layer of the films due to electrolyte/surface interaction, indicating a self-protecting mechanism. Nitride films showed the lowest values and less corrosion resistance, confirming the results obtained in polarization potentiodynamic tests. The coatings developed in this work, namely Cr and CrO, showed a promising corrosion resistance behavior that could endure a lifetime of frequent human touch in various decorative applications either automotive or general appliances.

## 1. Introduction

Traditionally, the method of giving a decorative coating to a substrate was through lacquering [1], which was proven to be harmful to the environment [2]. This process was gradually replaced by magnetron-sputtered coatings, which allow different colors to be deposited onto the surfaces [3]. Many of the surfaces in automotive industries are metal-coated plastic parts that replace metallic materials. Thermoplastics have a lower density and lower production costs. In terms of part design, this gives the manufacturers freedom as opposed to their metallic counterparts [4,5,6] while preserving the metallic look. However, not all automotive parts can be replaced by thermoplastics. The automotive industry struggles to find an equilibrium between the constant increase in weight due to increasing safety, luxury, and performance elements while complying with the need to increase fuel efficiency and emissions standards from European Union legislation (from <130 g CO_2_/km in 2015 down to <95 g CO_2_/km by 2021) [4,5]. 

Aluminum-based automotive components are still considered among the most promising materials to reach this equilibrium [6]. Combining a high-strength alloy with ease of part manufacturing makes aluminum ideal for safety and impact-absorbing components. Due to excellent corrosion resistance, aluminum and aluminum-based alloys can be used for cast engine blocks but more importantly for sheets and extruded parts like, hoods and trunk lids, and outer panels, such as doors, fenders, and protection covers [4,7,8,9].

Beyond the materials chosen, design, and manufacturing, different problems must be addressed in the automotive industry. The literature states that friction in the exterior and interior of automotives is a common cause of damage [10,11,12,13] and is often accompanied by corrosion. Corrosion is normally linked to metal oxidation of steels used in automotive bodies due to humidity, being the main cause of corrosion complaints in the industry [14]. Despite its excellent corrosion resistance in coated and uncoated conditions, aluminum, another common material in vehicles, can still develop galvanic, filiform, or crevice corrosion [15], which influences the end user’s perception of finishing quality. This aesthetic effect is more noticeable in regularly handled pieces such as doorhandles and buttons [9,10,11,12]. In automotive vehicles, corrosion can be due to rain and salt in the exterior, whereas in the interior it can be due to perspiration.

Human perspiration encounters many everyday objects, leading to unwanted effects such as discoloration, corrosion, and in the worst-case scenario malfunction due to the corrosive nature of sweat [16]. In an early attempt to understand the effects of corrosion on CrN/TiN decorative coatings for heavy human contact applications such as wristbands, Shimpi K. et al. [17] exposed coated stainless steel to supersaturated solution of sodium metabisulphite for porosity check and later analyzed samples for corrosion signs. The authors concluded that the CrN/TiN decorative coating protected the substrate from corrosion, but no real attempt to understand the perspiration effect on the coatings designed for wristbands was made.

Suo X. et al. [18], studied the corrosion properties of CrN coatings on aluminum alloy fabricated by magnetron sputtering through electrochemical techniques such as potentiodynamic measurement and salt spray tests, concluding that CrN coatings exhibited an excellent corrosion resistance on aluminum alloy, indicating that CrN could be a promising decorative and protective coating for equipment made of aluminum and exposed to corrosive environments, such as those addressed in the present paper. More recently, Abhijeet Yadav et al. [19] studied the correlation between sweat and hearing aid device failure. They found out that corrosive media such as human sweat was the primary cause for corrosion failures in those devices, hearing aids, where a higher corrosion rate was shown in the summertime. Studies on the effect of artificial sweat on PVD-coated and anodized materials for decorative applications were performed by Fenker et al. [20]. The authors immersed Nb-coated and anodized Ti samples in artificial sweat, mimicking the corrosion behavior of frequently touched colored objects such as eyeglass frames, cutlery, and others. They reported that corrosion can be prevented with a PVD coating.

We previously coated polycarbonate with Cr-based materials with different compositions (Cr, CrN, graded CrN, CrO) and morphology (multilayers) aiming to improve the hardness and allow a variation of the decorative appearance [21]. In this work, a follow-up on the previously referenced study, we aim to study the possibility of using the same coatings in different automotive components. Hence, we developed hexavalent chromium-free coatings onto aluminum substrates with an epoxy layer designed to enhance PVD coating adhesion. The substrate arrangement is often found in the automotive industry and can be used in appliances where a metallic finish is desired by the consumer. The coatings were deposited by reactive magnetron sputtering (a green technology) from a chromium target using nitrogen and oxygen as reactive atmospheres. Corrosion resistance against artificial sweat was tested to simulate the aggression the coatings would endure when touched during their lifetime.

## 2. Materials and Methods

Coatings were deposited by DC reactive magnetron sputtering in a home-made vacuum chamber onto monocrystalline silicon wafers (100 P-type) and polished Al plates (alloy AA 5052) samples. Aluminum substrates were protected with a transparent UV-cured basecoat (based on acrylate resins) before the metallization to improve both the adhesion and decorative appearance of the coating. 

Before being loaded into the vacuum chamber, samples were cleaned with isopropanol. During deposition, a single Cr target (Testbourne Ltd., Hampshire, UK, 99.5% purity) acting as the cathode was bombarded by Ar particles for 50 s, and a pure Cr adhesion layer deposited on the substrate created an interlayer. After, N_2_ and O_2_ as reactive gases were added at different flow rates as indicated in Table 1. By tuning the reactive gas flows, it is possible to change the appearance of the coatings, meeting end-user demand. Depositions were carried out at room temperature, and the chamber configuration allowed a constant 3.5 rpm rotation of the substrates. The target current density was kept at J_W_ = 20 mA·cm^−2^. The base pressure in the chamber was approximately 1 × 10^−3^ Pa, and the working pressure was kept approximately at 1 Pa. Before every deposition, an etching process was performed using a pulsed power source with 200 kHz, a pulse width of 1536 ns, and 400 mA current for 15 min. The deposition time of each set of samples is presented in Table 1. The multilayer coating (CrN/CrO) was deposited with similar conditions of monolithic coatings with each layer being deposited for 60s. For graded CrN coatings (gCrN), depositions were carried out with increasing nitrogen flow up to 25 sccm to match CrN conditions in the outermost portion of the coating.

The thickness and morphology of the coatings were analyzed using a NanoSEM FEI Nova 200, equipped with a Pegasus X4M for the EDS chemical composition analysis.

The structure of the coatings was analyzed by X-ray diffraction in a X′ Pert Pro MPD diffractometer operating with Cu Kα radiation (λ = 1.5406 Å at a grazing incidence angle of α = 3° and in a 2θ interval of 30–80°).

The adhesion of the coatings on aluminum substrates was tested using the cross-cut test following the ISO 2409 standard.

Surface roughness of the as-deposited coatings, and after 96 h of exposure to artificial sweat was assessed using atomic force microscopy (Icon Dimension, Bruker, Billerica, MA, USA) at room temperature with a conductive Si cantilever in contact mode. Three measurements were performed to evaluate average surface roughness (S_a_) and root-mean-square roughness (S_q_) in all samples according to the ISO 25178 standard and holding a 2.5 × 2.5 μm^2^ area.

The wettability studies were carried out using optical contact angle (CA) measurements (DataPhysics Instruments GmbH, Filderstadt, Germany, OCA20 equipment) with artificial sweat described above. Drop volume was maintained at <10 μL for each measure to avoid possible interferences in the contact angle measurements from the drop weight.

To evaluate the effect of sweat exposure on the coating’s surface, artificial sweat was made according to the industry standard TL 226 from the Volkswagen group. Ammonia, mass percentage w_(NH3)_ = 0.036%, and sodium chloride, w_(NaCl)_ = 0.5%, were added to water. In detail, 1 g of NH_3_ solution (25%, for analysis purposes) with 690.72 g of distilled water (in practice: 691 g) were used, and 3.47 g NaCl was added.

Corrosion behavior was induced via potentiodynamic polarization and electrochemical impedance spectroscopy (EIS) techniques, using artificial sweat to simulate long exposure to human touch for 96 h. All measurements were acquired in a Gamry 600 potentiostat connected to an electrochemical cell with a three-electrode configuration with the sample as working electrode (exposed area = 0.35 cm^2^), a platinum sheet as the counter electrode, and a saturated calomel electrode (SCE) as the reference electrode. The potentiodynamic polarization measurements were performed between −1500 mV and 1500 mV vs. E_ref_ with a 1 mV/s scan rate. EIS measurements were acquired at 10 points/decade, applying 10 mV rms, between 100 kHz and 0.01 Hz. All the samples were exposed for 3600 s to be stabilized before measurements. Three replicas of electrochemical tests were performed (for every coating in all exposure times). To fit the data from the potentiodynamic polarization test and calculate the corrosion rates, the ASTM G102-89 standard was followed.

## 3. Results

### 3.1. Chemical Composition (EDS)

The chemical composition (shown in Table 1) of coatings deposited onto Si wafer substrates was measured with EDS. All samples presented a high amount of oxygen due to a relatively high deposition pressure, of approximately 1 Pa, which is well known to promote oxygen incorporation on the coatings. This is true even for the Cr coating with an amount of O of 9 at. %. Such high working pressure was used due to the need to have coatings that can be simultaneously deposited onto polymeric substrates, for which we need a reduced adatom energy that prevents polymer degradation during sputtering [21].

For gCrN and CrN, nitrogen presence in the coatings was 28 at. % and 32 at. %, respectively. The CrN coating was made with a constant nitrogen flow, leading to a final higher amount of incorporated nitrogen.

CrO chemical composition showed that the reactive oxygen atmosphere promoted an oxygen-rich coating also promoted by the high reactivity of oxygen to Cr [22]. Notably, regarding the chemical composition of samples gCrN and CrN/CrO, for graded and multilayered samples, respectively, the composition is the combination or an average of the consecutive layers that make up the coatings and that was taken into consideration.

### 3.2. Structural Characterization (XRD)

The crystallographic structure was evaluated using X-ray diffraction (XRD), and the results are shown in Figure 1 for all deposited coatings.

For the chromium coating, Cr, an intense reflection close to 2θ = 44.4° the b.c.c α-Cr (ICCD card no. 01-085-1335) was observed.

On the gCrN XRD spectrum, it is possible to identify a broad peak between 42.0° and 44.5° that can be assigned to a mixture of different phases, ranging from Cr to CrN due to the graded composition. It is observed the b.c.c. α-Cr and reflection close to 2θ = 43.2° that corresponds to the hexagonal Cr_2_N phase ((−1 −1 1) plane from ICDD card no. 01-079-2159). We also observed CrN crystallographic phases.

For the CrN coating, deposited with a constant nitrogen partial pressure, a (111) peak from the CrN phase (ICCD card no. 01-076-2494) is observed. Near 2θ ≈ 39.4°, a strong peak corresponding to the diffraction of the (200) planes equidistant from the metastable Cr phase, δ-Cr, according to ICDD card no. 00-019-0323, is visible. A b.c.c. α-Cr phase close to 2θ ≈ 44.3° is explained by the sub-stoichiometric nature of the coating.

The sample CrO XRD pattern shows one diffraction peak between 2θ ≈ 42° and 47° which is composed of a main peak of α-Cr and a broad superposed peak that can be attributed to chromium oxide phases. On the Cr-O system, this peak at 2θ ≈ 43.6° can be a Cr_3_O phase (ICDD card no. 01-072-0528) or CrO0.87 phase (ICDD card no. 01-078-0722), both cubic structures. Due to tensile stresses, a peak shift to higher diffraction angles was observed.

On the multilayer coating, CrN/CrO, the analysis gives information from several layers of the coatings resulting in a mixture of the individual single-phase spectra of CrN and CrO.

The authors exhaustively discussed a more detailed in-depth analysis of similar samples on polymer substrates, which can be found elsewhere [21].

### 3.3. Morphology (SEM)

The thickness of the coatings was obtained by cross-section analysis from SEM micrographs on silicon substrate. Those results are shown in Table 1 and Figure 2. Surface morphology was also obtained from SEM micrographs.

Top-view SEM micrographs showed an increase in surface defects with the increase in nitrogen content, Figure 2. The introduction of oxygen as reactive gas for sample CrO and multilayer CrN/CrO promoted the appearance of large boulders on the surface. Since this morphology can be closely related to corrosion, a deeper analysis of morphology was conducted in correlation with corrosion behavior in Section 3.5 and Section 3.6.

Coating thickness ranged between 1100 nm and 1660 nm for CrN and CrO, respectively, and the deposition rate varied between 1.5 and 2.3 nm/s.

The deposition rate of Cr coating was 1.9 nm/s, and gCrN and CrN coatings showed a value of 1.6 and 1.5 nm/s. With the increase in the N_2_ flow ratio, the deposition rate decreased, which correlates with the consecutively lower thickness observed on samples Cr, gCrN, and CrN in Figure 2. When the partial pressure of the reactive gas increased inside the vacuum chamber, film growth behavior changed from a columnar-like type—usual in metallic sputtered Cr—to an increasingly denser columnar-like CrN coating, which indicates a lower deposition rate, as observed and reported by others [23].

The thickness increased for samples deposited with oxygen as reactive gas (CrO coating), and the deposition ratio reached 2.3 nm/s. This is a normal behavior of the Cr-O system [24].

For coating CrN/CrO, the deposition rate is the average of the deposition ratio of CrN and CrO.

The CrO micrograph showed the presence of consecutive layers either deficient or rich in O. The same behavior was observed on the CrN/CrO multilayer coating. This phenomenon is explained by deposition conditions and discussed by the authors elsewhere [21].

### 3.4. Adhesion (Tape Test)

Adhesion of the coatings was tested using the cross-cut tape test, and the results are presented in Figure 3.

According to the standard, the adhesion of the deposited coatings was rated 0 according to the ISO 2409 standard, meaning perfect adhesion. All coatings had perfect adhesion (Figure 3), where the edges of the cut were completely smooth, and none of the small squares that made up the grid were detached after removing the tape. The good adhesion behavior is also exponentiated by the acrylate layer between the coating and the substrate even for the CrN/CrO coating, whose surface seems cracked due to residual stress caused by the difference in coefficients of thermal expansion of the two materials in contact [25,26], but ultimately the adhesion of the coating was not affected.

### 3.5. Roughness and Wettability

The roughness of the coatings was measured as deposited and after corrosion test conditions to assess the damage caused by the artificial sweat over the films. Table 2 and Figure 4 and summarize the results obtained from AFM measurements.

Concerning the films, Cr exhibited a negative skewness with low kurtosis, which means some low bumpy holes in the analyzed area (see Figure 4). After corrosion tests, this surface showed some spiky mounds maintaining the presence of holes. Its increase in S_ku_ (from 0.4 to 2.3) and decrease in S_sk_ (from −0.7 to −1.2) indicate the increase in valleys in the film after sweat interaction but with fewer height differences as demonstrated by the height bar in Figure 4 (Cr after corrosion tests). Some particles over the film surface were noted after corrosion tests in this sample, which could be related to a possible oxidation process. It was quite surprising to note that Cr film did not exhibit important harm after the corrosion test.

Apparently, the variations in areal roughness were strongly influenced by the electrolyte in nitride films (gCrN and CrN), exhibiting ~10 times higher S_q_ values, skewness (S_sk_)—which denotes the spikes (positive values) or valleys (negative values) present on the analyzed area [27]—and kurtosis (S_ku_)—which qualifies the width of the height distribution [28] or the sharpness of the height profile [27,29]. According to these definitions, the decrease in S_sk_ could be related to the presence of pitting in the sample surfaces after corrosion tests (see Figure 5).

Furthermore, the nitride films (gCrN and CrN samples) displayed a strong variation of skewness, achieving more negative values after corrosion tests. Figure 5 exhibits that nitride films suffer an aggressive pitting process, which could be considered the corrosion mechanism in these samples. This behavior could be promoted by defects in the film (i.e., holes), which could facilitate the electrolyte sinking, and corroding during the exposure time (96 h), like the findings of Chipatecua et al. [30] in CrN/Cr multilayers. Indeed, the nitride films displayed the highest increase in roughness (S_a_ and S_q_) after corrosion tests.

CrO was not apparently affected by the artificial sweat after 96 h (see Figure 4 and Figure 5). This sample shows a slight increase in S_ku_ and S_sk_ (from 0.1 to 1.4 and 0.2 to 0.7, respectively), but this did not influence the film integrity.

On the other hand, CrN/CrO displayed a S_ku_ decrease (from 1.0 to −0.2), denoting that the “defects” in the film surface are blunter, reflected in large corroded areas (at a deeper level) with other less corroded areas (see Figure 5), hence suggesting an uniform corrosion action.

The nitride samples (CrN and gCrN) seem to show the most corroded surfaces when exposed to artificial sweat. Based on AFM results, film defects have a determinant role in triggering the infiltration of the electrolyte into the film, prompting its posterior detachment. Nonetheless, this action was more controlled in the multilayered CrO film because the oxide layer acted as a “brake” against the chemical action of the electrolyte, demonstrating immunity just like the CrO film.

Wettability measurements with artificial sweat were performed to assess the capacity of the film to allow the electrolyte sinking. According to the values exhibited in Table 2, nitride films (mainly the CrN sample) displayed less hydrophobic behavior than oxides films. This behavior was influenced by the nature of the film surface as aforementioned. Remarkably, Cr film showed the highest contact angle value with artificial sweat, which helps to explain its relative stability, displaying fewer changes in roughness and morphology after corrosion tests, despite being a high-purity metallic film. The obtained films could be coherent with the Cassie–Baxter model [31]. Some studies have demonstrated that hydrophobicity is a critical surface characteristic to avoid corrosion, as mentioned by Jin et al. [32]. Other authors related the density and absence of superficial defects on films to avoiding corrosion events [33,34]. According to our findings, the phase composition (Cr + CrN), the nanocrystalline structure, and the less hydrophobicity displayed for nitride films played the main role in explaining the film degradation. It is possible that a chemical reaction between CrN and the electrolyte could affect the film performance too as shown by Ibrahim et al. [35], who measured the corrosion resistance of CrN films against 0.5 M NaCl and borate buffer (pH 9.0) solutions. The CrN film exhibited worse anticorrosive properties whilst exposed to the borate solution. The authors explained this action as the consequence of two combined factors: the solution capacity to dissolve the film and the corrosion process through pinholes in the film morphology.

### 3.6. Corrosion Resistance against Artificial Sweat

Figure 6 exhibits the potentiodynamic curves of all samples and the substrate with base coat. According to the potentiodynamic polarization tests, all coatings displayed a nobler behavior than the substrate. All the sputtered coatings showed lower corrosion current densities than the substrate. As a first insight, all the samples showed two regions of passivation with their perspective breakdown zones. Just Cr showed a quick passivation, and afterwards the current density always increased constantly, which suggests that the anodic reaction did not come close to stopping at any time. On the other hand, gCrN exhibited a passivation of the surface and its respective breakdown with the increase in potential. In fact, gCrN displayed several current fluctuations in the second passivation zone (above +0.5 mV vs. E_ref_), which indicates the presence of pitting events during the tests and confirms the pits exhibited in Figure 5, as discussed before. After the second passivation, the transpassive zone always appeared, showing stable behavior in terms of current densities.

Current densities measured by Tafel extrapolations are strongly associated with the polarization resistance (*R_p_*). According to the ASTM G3 standard, *R_p_* could be calculated from [36]:(1)$$R_{p}=\frac{\beta_{a}*\beta_{c}}{2.303*\left(\beta_{a}+\beta_{c}\right)*j_{corr}}$$
where *β_a_* and *β_c_* are the anodic and cathodic Tafel slopes, respectively, and *j*_corr_ is the corrosion current density. The fit parameters as well as every calculated R_p_ is displayed by Table 3. *R_p_* values demonstrated that nitride films displayed the worst anticorrosive behavior against artificial sweat. The presence of nanocrystalline structure and the phase composition compromise the corrosion behavior. This approach also justifies the behavior of the multilayer system. In fact, the nitride phase affects the corrosion performance of multi-layered CrN/CrO film, which shows closer values of *J_corr_* to CrN film and exhibits uniform corrosion (see Figure 5). The CrO film showed the highest polarization resistance, indicating higher corrosion resistance against artificial sweat. Our results are coherent according to the description by Dinu et al. [37], who studied the corrosion behavior of chromium-based films. They produced CrN, Cr(N,O), Cr(N,O)/CrN, and CrN/Cr(N,O) coatings over 304 stainless steel coupons. Their findings pointed out the crucial role of oxygen in the outer film layer leading to higher *R_p_* values and consequently better corrosion resistance.

Electrochemical impedance spectroscopy (EIS) results pointed out to a diminution of impedance modulus (|Z|) after 96 h systematic immersion, which denotes a diminution of corrosion resistance against artificial sweat of the obtained films (see Figure 7). Nitride films consistently exhibited the worst anticorrosive properties. On the other hand, Cr and CrO exhibited the highest |Z| values, meaning the best corrosion resistance, along with corrosion rate forecasts (see Table 3). The multilayered CrN/CrO sample could not be used to calculate the corrosion rate owing to the difficulty of estimating or assuming its density with a certain reliability. Also, the phase angle plots showed the largest differences at low frequencies (<10 Hz), which exhibited an increasing trend in angles with higher exposure time.

Additionally, Nyquist plots (Figure 8) evidenced the capacitive nature of the films. At 96 h of exposure to the artificial sweat, the Nyquist plots evidenced the strong shrinking of initial semicircles shown by the films. Cr and CrO films displayed light steeping to low frequencies, which could indicate some degradation of the capacitive behavior. Nitrides and multi-layered CrN/CrO samples kept the same initial shape in Nyquist plots. All these measurements were adjusted according to the equivalent electrical circuit (EEC) exhibited in Figure 9.

Regarding the EEC, the circuit is composed of five elements. R_sol_ represents the resistance of artificial sweat. Furthermore, R_coat_ and CPE_coat_ represent the coating’s resistance and constant phase element, respectively. Commonly, CPEs are used to reproduce the electrochemical response of the surface when it is not working ideally. Defects like pores, pinholes, and heterogeneities in topography or chemical species along the surface (i.e., impurities) are common in sputtered films. These elements are classified according to a variable exponent α, which can vary between 1 and 0. When this exponent is closer to 1, the CPE behaves as a capacitor; between 0.6 and 0.4, it is considered as a resistor connected in series with a capacitor (or a Warburg element); and 0 means that the CPE works like a resistor [38]. Lastly, R_p_ and CPE_dl_ represent the charge transfer resistance and the double-layer capacitance, respectively. The latter is related to the substrate/coating interaction. Table 4 shows the highest R_p_ values in Cr and CrO for the obtained films. In fact, R_p_ value showed by Cr (~2.2 MΩ), the closest one to CrO (~7.8 MΩ), confirms the formation of the oxides in these films as the electrolyte/film interaction, and, hence, this sample has a self-protecting mechanism (passive layer) as was mentioned before. In contrast, nitride films showed the lowest values and, hence, less corrosion resistance, confirming the results obtained in polarization potentiodynamic tests. Another important aspect is the α exponent of CPEs. CrO was the only one which showed α value (~0.52) pointing to the CPE_coat_ behaving like a Warburg element (W), meaning that a slight diffusion process occurred but was not significant enough to replace the CPE for a W in the EEC [39]. The remaining α values in CPE_coat_ were between 0.66 and 0.84; hence, the remaining films were demonstrated to be far from the ideal capacitive behavior. These values are consistent with surfaces with a high grade of defects like voids, such as SEM and AFM results (see Figure 4 and Figure 5). On the other hand, α values in CPE_dl_ were 0.62 and 0.79. These values reflect the substrates’ high roughness and surface defects (Aluminum with UV-cured basecoat), expected in a painted metallic surface (Appendix A). Fit parameters showed chi-square values in the magnitude of 10^−3,^ which could be considered a good fit [40]. All other fit parameters are shown in Table 4.

EDS measurements were performed after corrosion tests to verify the chemical changes in the samples after artificial sweat interaction. Figure 10 exhibits chemical composition and the O/Cr ratio between as-deposited and corroded conditions from all obtained films. The artificial sweat significantly oxidized the film surfaces, confirmed by the increase in all O/Cr ratios. The formation of oxides in some areas is probably due to the heterogeneity of the films, as mentioned before, which is exhibited in Figure 5. The chemical kinetics of film after the electrolyte interaction was not measured in this study. However, some studies point out the possibility of CrN oxidation. Indeed, Gao et al. [41] established that the CrN films could oxidize in some hydroxide and oxide species. According to their XPS results after 4 h of electrolyte exposure (3 h in a buffer solution and 1 h in a 3.5 wt.% NaCl + buffer solution), CrN displayed strong superficial passivation with Cr_2_O_3_, Cr(OH)_3,_ and CrO_3_ species. The formation of Cr_2_O_3_ is plausible in the obtained films according to the EDS results after the corrosion tests, which exhibited a quasi-direct exchange between Cr and O without affecting the N content in the film. This phenomenon is possible due to the preferential formation of this oxide with free Cr in the film structure as is demonstrated by the Cr-O phase diagram [42] and the enthalpy of formation of these compounds (Δ_f_H°_CrN_ = −117.15 KJ/mol & Δ_f_H°_Cr2O3_ = −1134.70 KJ/mol [43]). Also, the EDS results before corrosion tests exhibited the substoichiometric conditions of the films, hence the possibility of having Cr free atoms which could react with an oxidizing media as the artificial sweat. It is possible that a mixture of free Cr crystals and CrN in the film’s structure would be affected by the electrolyte, leading to an oxidation process promoted by OH^−^ and the dissolution of this passive layer provoked by the presence of Cl^−^ ions as proposed Gao et al. [41,44].

We further investigated the possible reactions involved at the coating–electrolyte interface for a better understanding of the chromium nitride morphology after corrosion tests.

In the presence of aqueous NH_3_, a high concentration of OH^−^ ions will be present in the electrolyte (pH = 11.7), Equation (2):NH_3 (aq_._)_ + H_2_O _(l)_ → NH_4_^+^ _(aq)_ + OH^−^ _(aq_._)_(2)

Through EDS analysis, it is possible to see an increase in the oxygen content of the samples after corrosion tests. By naked eye visualization (Appendix A), it is possible to observe the coverage of CrN by a dark layer that can indicate the presence of chromium oxide. This allows us to identify the probable reaction pathway involved in the reduction–oxidation mechanism of CrN to Cr_2_O_3_ (Equation (3)) because the coatings will gradually oxidize to the thermodynamically stable oxides.
2 CrN _(s)_ + 3 OH^−^ _(aq_._)_ → Cr_2_O_3 (s)_ + N_2 (g)_ + 3 H^+^ _(aq_._)_(3)

The pitting in the chromium nitride samples observed by SEM analysis after electrochemical tests (Figure 5) can be due to the release of molecular nitrogen from the coating to the electrolyte. In the presence of a synthetic sweat electrolyte, nitrogen is released into the solution and does not remain trapped within the solid layer.

## 4. Conclusions

In an effort to develop hexavalent chromium-free coatings for frequently touched automotive parts, Cr(N,O) and multilayered CrN/CrO coatings were successfully deposited on aluminum substrates by reactive magnetron sputtering.

The chemical composition showed that all coatings had an excess of oxygen that can be related to high deposition pressure. chemical and structural characterization of gCrN and CrN/CrO multilayered coatings showed a combination of the monolithic coating’s characteristics.

Morphologically, Cr coating showed a columnar structure that turned denser with the addition of N_2_ as a reactive gas. When oxygen was added as reactive gas (CrO coating), the coating increased in thickness, and, due to the deposition chamber setup, repeated layers of either O-deficient or O-rich material were deposited. Once again, CrN/CrO coating shows a mixture of both CrN and CrO morphology throughout the coating.

Regarding coating stability on the epoxy-coated aluminum substrates, all samples presented good adhesion to the substrates enhanced by the epoxy used.

Corrosion behavior was induced using artificial sweat to simulate long exposure to human touch for 96 h. In potentiodynamic polarization tests, the coatings showed nobler behavior then the naked substrate. Cr almost does not possess a passivation region. Meanwhile, gCrN exhibited a passivation of the surface and its respective breakdown with several current fluctuations above +0.5 mV vs. E_ref_, indicating the occurrence of pitting, which was confirmed by SEM micrography after the corrosion tests. The remaining films displayed two zones of passivation with their respective breakdown regions. Regarding EIS results, all films depicted a diminution of impedance modulus (|Z|) after 96 h, indicating a diminution of corrosion resistance against artificial sweat. Nitride films exhibited the worst anticorrosive features governed by film defects that cause electrolyte sinking. On the other hand, Cr and CrO exhibited the highest |Z| values. These results are corroborated by low the corrosion rates of both coatings.

Equivalent electrical circuits of obtained coatings are composed of five elements: R_sol_, R_coat_, CPE_coat_, R_p_, and CPE_dl_. R_p_, the charge transfer resistance was higher for Cr (~2.2 MΩ) and CrO (~7.8 MΩ), confirming oxide formation due to electrolyte/surface interaction and indicating a self-protecting mechanism in line with previous assessments. On the other hand, nitride films showed the lowest values and less corrosion resistance, confirming the results obtained in polarization potentiodynamic tests.

The coatings developed in this work, namely Cr and CrO, showed promising behavior that could endure a lifetime of frequent human touch in various applications, either automotive or general appliances.

## Figures and Tables

**Figure 1 nanomaterials-13-02346-f001:**
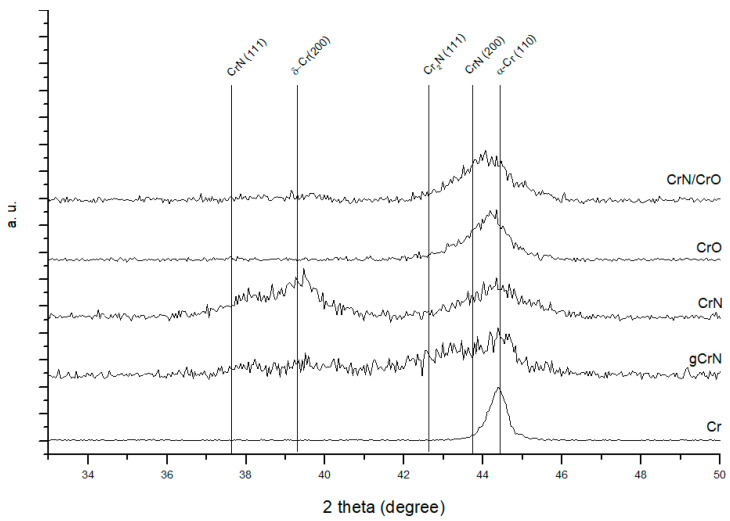
XRD patterns of the deposited coatings.

**Figure 2 nanomaterials-13-02346-f002:**
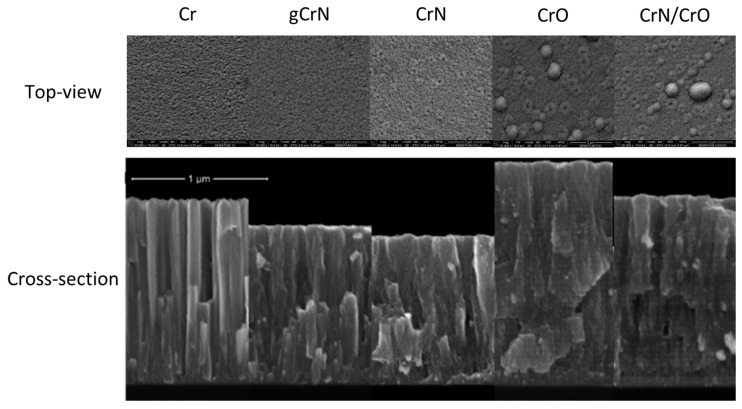
Scanning electron top-view and cross-section micrographs of the coatings deposited on silicon.

**Figure 3 nanomaterials-13-02346-f003:**
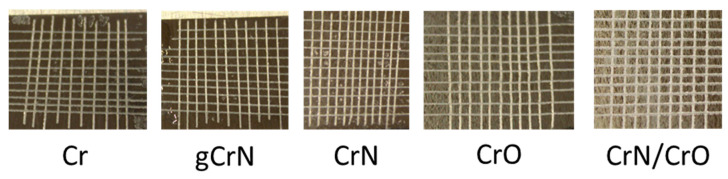
Adhesion behavior assessment of coatings deposited onto aluminum substrates.

**Figure 4 nanomaterials-13-02346-f004:**
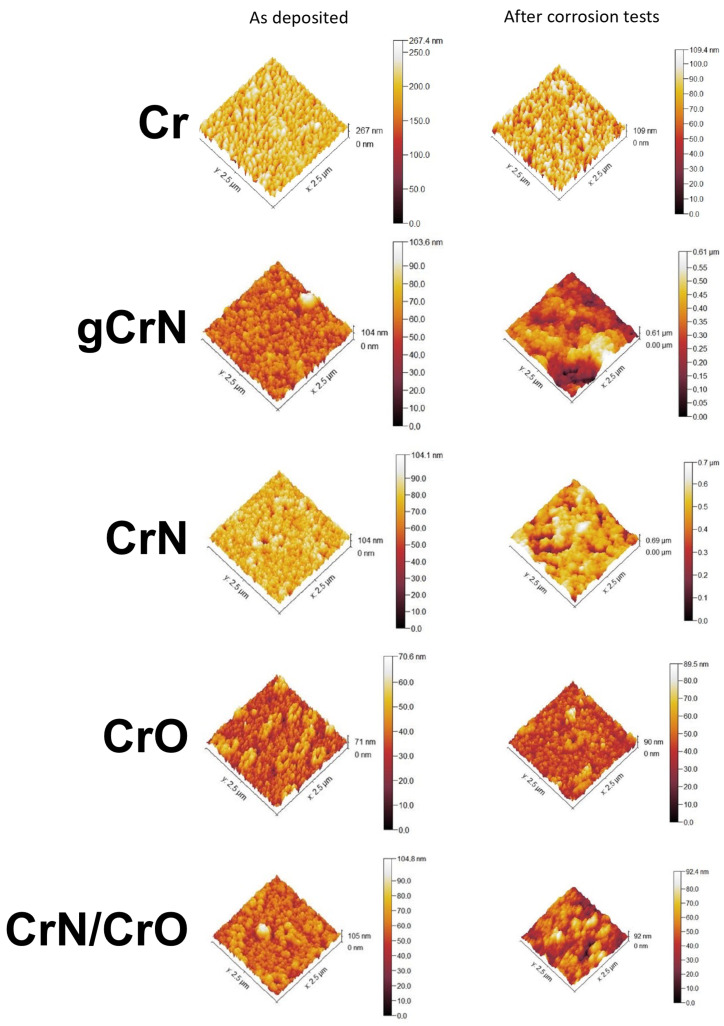
AFM measurements from obtained coatings in as-deposited conditions and after corrosion test (96 h of artificial sweat exposure).

**Figure 5 nanomaterials-13-02346-f005:**
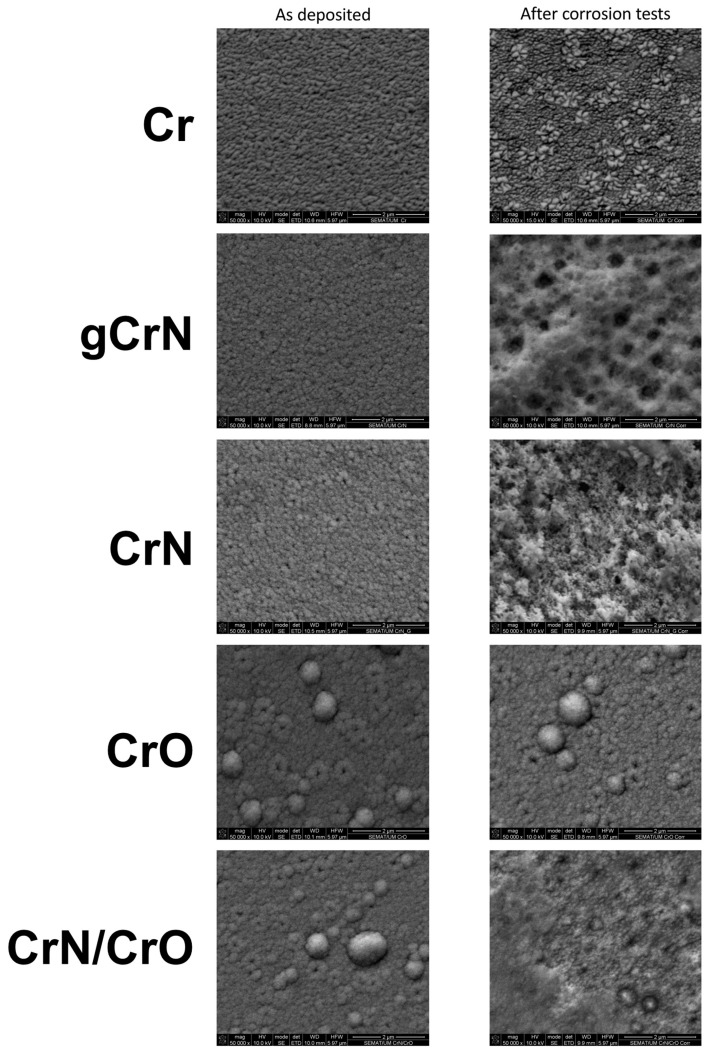
SEM micrographs from coatings in as-deposited condition and after corrosion test (96 h of artificial sweat exposure).

**Figure 6 nanomaterials-13-02346-f006:**
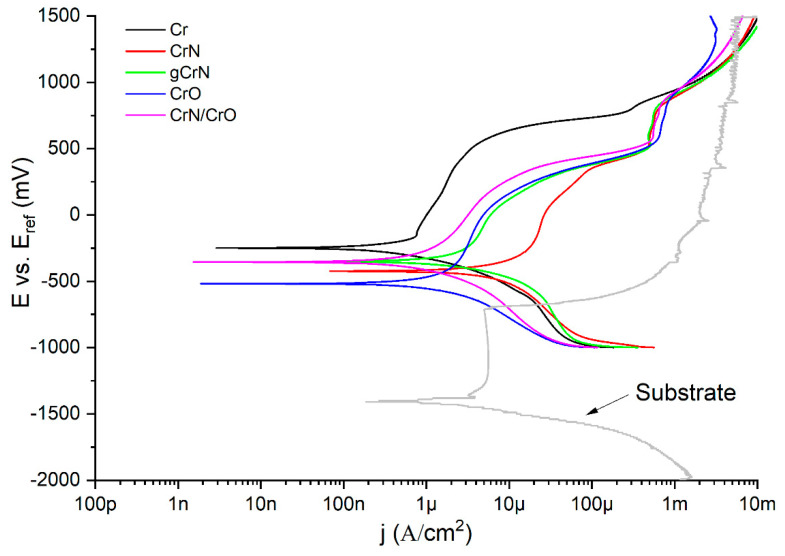
Tafel plots from obtained films and substrate.

**Figure 7 nanomaterials-13-02346-f007:**
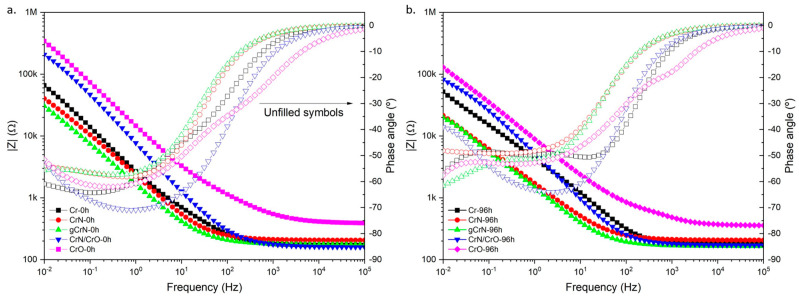
Bode plots of obtained coatings to 0 h (**a**) and 96 h (**b**).

**Figure 8 nanomaterials-13-02346-f008:**
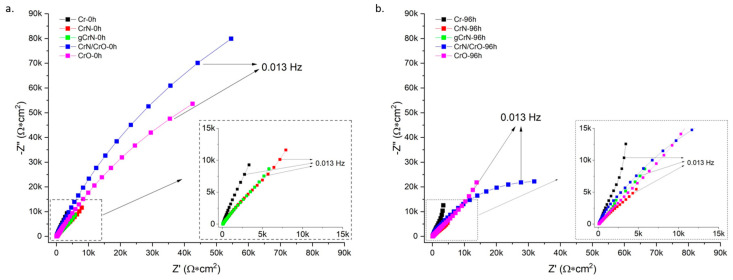
Nyquist plots of obtained coatings to 0 h (**a**) and 96 h (**b**).

**Figure 9 nanomaterials-13-02346-f009:**
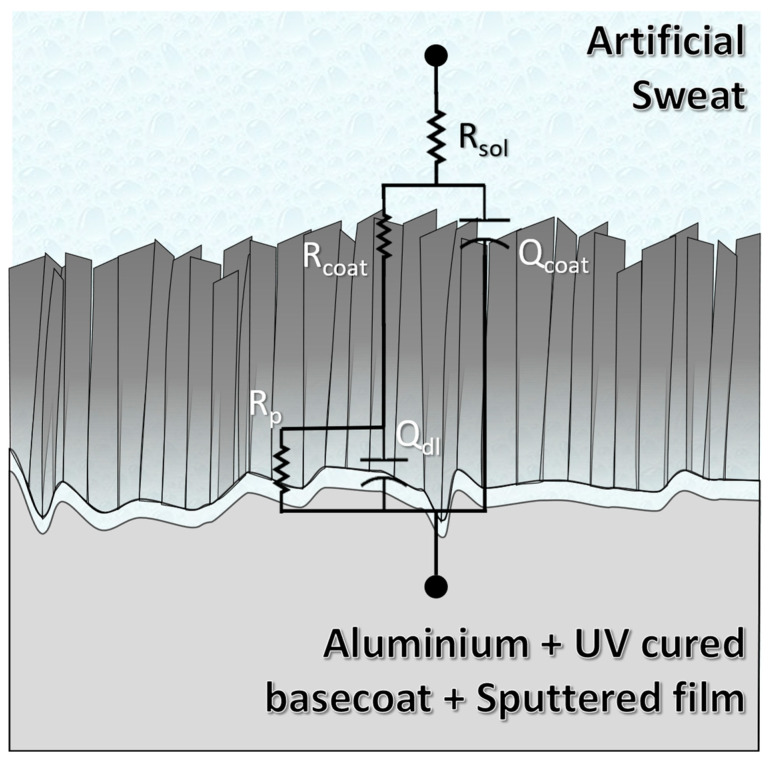
Equivalent electrical circuits of obtained coatings.

**Figure 10 nanomaterials-13-02346-f010:**
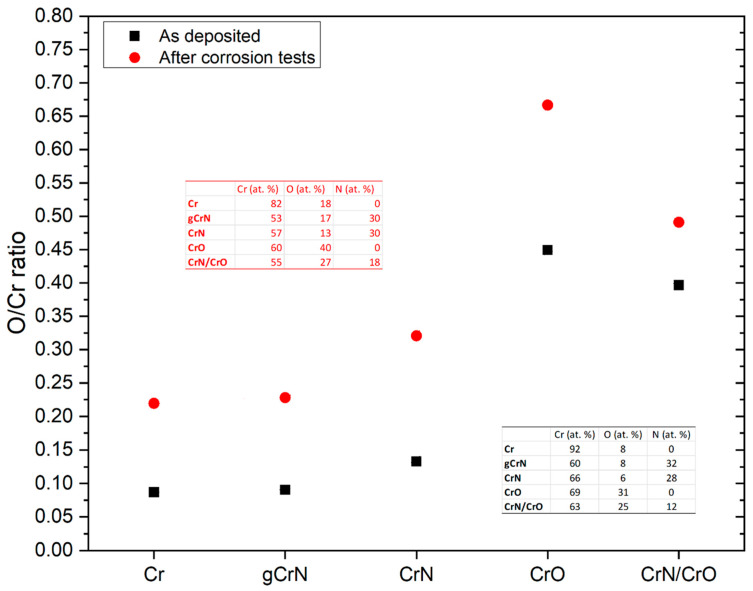
O/Cr ratio and chemical composition of the obtained films before and after corrosion tests.

**Table 1 nanomaterials-13-02346-t001:** Deposition conditions of the coatings (gas flow and deposition times), deposition rate, and chemical composition characterization of the deposited coatings.

Coating	Gas Flow (sccm)			Deposition Time (s)		Thickness	Deposition Rate	Chemical Composition (at. %)		
	Ar	N_2_	O_2_	Interlayer	Coating	(nm)	(nm/s)	Cr	O	N
Cr	100				720	1390	1.9	91	9	-
gCrN	70	5→25			720	1180	1.6	66	6	28
CrN	70	25		50	670	1110	1.5	60	8	32
CrO	100		15	50	670	1660	2.3	69	31	-
CrN/CrO	70	25	15	50	720	1420	1.8	63	25	12

**Table 2 nanomaterials-13-02346-t002:** Roughness and contact angles (measured with artificial sweat on as-deposited films) from obtained coatings.

Sample	As Deposited			After Corrosion Tests			Contact Angle (°)
	*S_q_* (nm)	*S_sk_*	*S_ku_*	*S_q_* (nm)	*S_sk_*	*S_ku_*	
Cr	14.4 ± 0.1	−0.7	0.4	27.0 ± 0.3	−1.2	2.3	122.6 ± 1
gCrN	9.5 ± 0.2	0.1	2.1	93.7 ± 4.2	−1.1	2.3	105.2 ± 1
CrN	8.6 ± 0.1	−0.6	1.9	92.8 ± 3.8	−1.1	1.5	94.6 ± 1
CrO	8.3 ± 0.1	0.2	0.1	8.4 ± 0.2	0.7	1.4	108.5 ± 2
CrN/CrO	9.8 ± 0.1	0.2	1.0	12.9 ± 0.5	0.1	−0.2	108.1 ± 4

**Table 3 nanomaterials-13-02346-t003:** Fit parameters of Tafel plots and polarization resistances from coatings.

	β_A_ (mV/decade)	|β_C_|(mV/decade)	J_corr_ (nA/cm^2^)	E_corr_ (mV)	R_p_ (KΩ*cm^2^) *	Corrosion Rate (×10^−3^ mm/yr)
Cr	170 ± 58	193 ± 18	476 ± 67	−238 ± 11	82	5
gCrN	124 ± 7	309 ± 63	3471 ± 129	−258 ± 92	11	27
CrN	203 ± 38	139 ± 14	5075 ± 1335	−409 ± 15	7	35
CrO	127 ± 26	74 ± 15	321 ± 23	−505 ± 14	63	3
CrN/CrO	195 ± 30	94 ± 13	486 ± 19	−364 ± 7	57	-

* Substrate: R_p_ = 16.73 KΩ*cm^2^.

**Table 4 nanomaterials-13-02346-t004:** EIS fit parameters of obtained coatings.

	R_sol_ (Ω)	R_coat_ (Ω)	CPE_coat_ (µS*S^α^)	α_coat_	R_p_ (kΩ)	CPE_dl_ (µS*S^α^)	α_dl_	χ^2^ (×10^−4^)
Cr	173 + 2	671 + 65	46 + 12	0.82 + 0.02	2205 + 1441	124 + 20	0.69 + 0.03	2.5
gCrN	167 + 1	4899 + 2450	267 + 55	0.73 + 0.02	190 + 77	169 + 41	0.72 + 0.06	1.8
CrN	198 + 2	3286 + 1092	314 + 94	0.66 + 0.04	44 + 17	322 + 36	0.75 + 0.03	23.8
CrO	195 + 3	3844 + 1166	23 + 7	0.54 + 0.05	7829 + 2660	9 + 2	0.79 + 0.02	25.6
CrN/CrO	167 + 5	20685 + 3535	31 + 4	0.84 + 0.03	275 + 195	24 + 11	0.62 + 0.02	7.6

## Data Availability

Not applicable.

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
