# Peer review of "Corrosion Resistance in Artificial Perspiration of Cr-Based Decorative Coatings"

_nanomaterials, 2023, doi:10.3390/nano13162346_

Round 1
Reviewer 1 Report
In this work, Cr-based coatings were fabricated on modified aluminum substrates using DC reactive magnetron sputtering for inhibition the sweat corrosion in automobile industry. In general, authors conducted an integrated research using the correct techniques; the manuscript is well-organized and written that can be reconsidered after dealing with the following issues.
--- I cannot fine the definition of item "gCrN", miss it? At least its definition should be presented in the first emergence in Abstract Section.
--- In the Introduction Section, paragraphs 3 and 4 can be merged for concise, the same as paragraphs 6, 7 and 8. In addition, the literatures survey on corrosion and protection for devices in automobiles should be improved, and make a wider coverage on typical metals (e.g., Zn-Mg-Al coated steel, copper, referring 10.1016/j.corsci.2022.110957) and typical corrosion forms.
--- The type, raw materials and coating procedure of basecoat for aluminum should be described in detail, which are the key parameters to explain the favorable adhesion and strength for Cr-based coatings.
--- Pronounced noises were detected in XRD patterns of gCrN, CrN and CrN/CrO. In general, after the film was sputtered on the heterogenous substrate, typical lattice planes can be shifted. This is evidenced by the patterns of Cr and CrO. Thus, the shift of other patterns can be well-distinguished.
--- For the decorative purpose, the optical morphologies of different coated specimens should be provided in the fresh and corroded states (authors can refer to 10.1016/j.matlet.2023.133979). The largest retention of surface lustrous is the other key parameters that must be considered in the application of decorative coating.
--- For the deposited metallic coatings, is tape test proper for obtaining the adhesion strength?
--- In Table 3, the value of βa for the specimen with CrO coating attains 1493 mV/dec, which is an extremely large for the anodic branch. That means that it is impossible for anodic dissolution. However, this is hardly reflected by the corresponding curve. The same as βa values of 800 and 416 mV/dec. Please check them. In addition, the unit of corrosion rate is absent.
--- Several stylistic errors should be checked throughout the context. For instance, line 45, "[4][5]" is not standard the same as other analogue places; the line spaces may be another problem …
Reviewer 2 Report
The quality of this work is " technical ". Science does not belong to the objectives and aims of the Authors. But this is not surprising , since in the "Acknowledgements " they clearly wrote : "This work was supported by COMPETE 2020 a Portuguese and European Union initiative through the Project POCI-554 01-0247-FEDER-072607, R&D and production of logos for the automotive industry."
We aim to develop.... to be changed in " We aim at developing...."
line 112 ..... at different flow ratesas indicated in....... ..... at different flow rates as indicated in.......
line 179. The crystallographic structure was evaluated using X-ray diffraction (XRD), and....... The crystallography was evaluated using X-ray powder diffraction (XRPD), and.......
line 194. a strong peak corresponds to the reflection of the (200) plane...... a strong peak corresponds to the diffraction of the (200) planes equidistance.....
Round 2
Reviewer 1 Report
The manuscript has been revised.